# Effects of Non-Pharmacological Sleep Interventions in Older Adults: A Systematic Review and Meta-Analysis

**DOI:** 10.3390/ijerph20043101

**Published:** 2023-02-10

**Authors:** Hye-Ja Gu, Oi-Sun Lee

**Affiliations:** 1Department of Nursing Science, Kyungsung University, Busan 48434, Republic of Korea; 2Department of Nursing, Gyeongnam Geochang University, Geochang-gun 50147, Republic of Korea

**Keywords:** meta-analysis, non-pharmaceutical intervention, older people, review, sleep

## Abstract

This study investigated the effects of non-pharmacological interventions on sleep in older people through a systematic review and meta-analysis. We conducted a literature search using eight electronic databases according to the Preferred Reporting Items for Systematic Reviews and Meta-Analyses protocol. Participant characteristics, the contents of the evaluated interventions, and the measured outcomes were systematically reviewed for 15 selected studies. We performed a meta-analysis to estimate the effect size for overall, aggregated sleep outcomes. Due to the small number of studies available for each intervention, only the overall effectiveness of non-pharmacological sleep interventions was evaluated. The evaluated interventions included exercise, aromatherapy, acupressure, cognitive behavior therapy, and meditation. Our results demonstrated that non-pharmacological interventions showed statistically significant effects on sleep (effect size = 1.00, 95% confidence interval: 0.16, 1.85, I^2^ = 92%, *p* < 0.001). After confirming publication bias and removing outliers, we found no heterogeneity (I^2^ = 17%, *p* = 0.298), with a decrease in effect size to 0.70 (95% confidence interval: 0.47, 0.93). Non-pharmacological interventions are effective for improving sleep in older adults. Future studies should continue to investigate sleep problems and interventions addressing these problems in this demographic, particularly in older women. Objective measures should be used to follow-up on the evaluated sleep interventions over the long term.

## 1. Introduction

Sleep is a basic need for humans to maintain health and well-being [1]. Adequate sleep rejuvenates the body and plays a critical role in cognitive function, memory, and emotional regulation [1]. Conversely, sleep problems impact physical and emotional well-being as well as the immune system [2]. Chronic sleep deprivation and sleep disorders are key public health issues that lead to different health problems [3].

Sleep problems commonly occur among older adults [4], as changes experienced during the aging process influence sleep duration and quality [5]. Aging leads to diminished slow wave activity (SWA) and increased sleep fragmentation, and poor sleep damages memory capacity [6]. In addition, poor sleep alters sleep patterns, aggravates any existing cognitive problems, and reduces quality of life [7]. Sleep problems in older adults increase the risk of cardiac metabolic disorders [8]; Alzheimer’s disease [9]; and frailty, morbidity, and mortality [10]. These findings highlight the critical need to devote more resources to gaining a comprehensive understanding of sleep problems and interventions in older adults.

Various pharmacological and non-pharmacological interventions are routinely employed to address sleep problems. Non-pharmacological interventions are particularly attractive to the general population because these interventions are perceived as not causing adverse effects [11]. Interventions such as exercise [12], aromatherapy [13], auricular acupressure [14,15], cognitive behavioral therapy [16,17], meditation [18,19], and stimulation therapy [20] have been validated and well utilized. However, sleep problems remain a substantial health issue and a highly prevalent health problem with unclear guidelines as to effective management [21]. This calls for a systematic review to comprehensively examine the characteristics and effects of non-pharmacological sleep interventions in older adults.

Among Asian countries, Japan, Hong Kong, Taiwan, and the Republic of Korea have the highest proportions of individuals aged ≥65 years; these proportions are projected to increase substantially by 2060 [22]. Sleep is critical to health and quality of life in older adults, and sleep disorders are linked to many unfavorable medical states [23]. In the Republic of Korea, a recent systematic review evaluated the effects of non-pharmacological interventions in older adults [24]. Some meta-analyses have been performed on the effects of specific interventions, including aromatherapy [25,26], exercise [27,28], meditation [3,29], and auricular acupressure [30]. To our knowledge, no previous systematic reviews globally have formally analyzed the effects of non-pharmacological interventions on sleep in older adults.

In this context, we comprehensively examined the findings of experimental epidemiologic studies evaluating non-pharmacological sleep interventions in older adults to present scientific evidence for developing medical guidelines for effective sleep interventions. Our specific objectives were to analyze the characteristics of non-pharmacological sleep intervention studies, evaluate the methodological quality of non-pharmacological sleep intervention studies, formally analyze the effects of non-pharmacological sleep interventions, and evaluate potential publication bias in studies conducted among older adults.

## 2. Materials and Methods

### 2.1. Study Design

We conducted a systematic review and meta-analysis evaluating the effects of non-pharmacological sleep interventions in older adults.

To answer our key questions, we established the following study inclusion and exclusion criteria as per the Preferred Reporting Items for Systematic Reviews and Meta-Analyses (PRISMA) 2020 guidelines [31]. This review was neither registered, nor a protocol was prepared. The participants, intervention, comparison, outcome, and study design (PICO-SD) strategy for study selection was applied as follows:

(1) Participants: Older adults aged ≥65 years. Studies including both younger and older adults as well as studies that did not include older adults were excluded.

(2) Intervention: Non-pharmacological sleep interventions in older adults. Studies involving the provision of pharmacological interventions or non-provision of non-pharmacological interventions were excluded.

(3) Comparison: One-group studies without a control group were excluded.

(4) Outcomes: Outcome measures were not specified. All sleep-related outcome measures (positive and negative) were included. Studies for which the effect size could not be calculated were excluded.

(5) Study design: Randomized controlled trials (RCTs) published in peer-reviewed journals were included. Non-RCTs, experimental studies without a control group, pilot experimental studies, protocols, theses/dissertations, qualitative studies, descriptive surveys, reviews, and systematic reviews/meta-analyses were excluded. Studies without full texts (e.g., published abstracts, poster presentations, and conference proceedings) were excluded.

### 2.2. Search Strategy

A simulation search on PubMed for non-pharmacological sleep intervention studies in older adults showed that 83% of the search results were studies published since 2010 (final search date: 22 March 2022). Hence, the search period was set from January 2010 to March 2022. The language filter was set to Korean or English. Literature searches were performed in the MEDLINE, PubMed, Cochrane Library, CINAHL, NDSL, KISS, RISS, and DBpia databases. Searches were performed independently by two researchers. Disagreements on study inclusion were resolved by consensus.

After confirming that exercise, aromatherapy, cognitive behavioral therapy, and meditation interventions were applied as non-pharmacological sleep interventions in the simulation search, we designed our English-language database search strategy to include these studies. We used MeSH terms and natural language, along with truncation and phrase searches. The search terms were: “aged” (MeSH), “elderly,” “older adult,” “elder,” “older,” “sleep” (MeSH), “insomnia” (MeSH), “sleep disorders, intrinsic” (MeSH), “cognitive behaviour therapy” (CBT; MeSH), “exercise therapy” (MeSH), “aromatherapy” (MeSH), “meditation” (MeSH), “mind-body therapies” (MeSH), “sleep program,” “sleep intervention,” and “sleep therapy.” The search strategies were as follows: (1) aged OR elder* OR older*; (2) sleep* OR “sleep disorder*” OR insomnia; (3) “cognitive behaviour therap*” OR “exercise therap*” OR “aromatherapy” OR meditation or “mind body therap*” OR “sleep program*” OR “sleep intervention*” OR “sleep therap**”; and (4) (1) AND (2) AND (3).

Based on these search strategies and those employed in a meta-analysis of adults with sleep disorders in Korea [32], the key words for Korean search databases were set to: “older adults,” “sleep,” “insomnia,” “therapy,” “intervention,” and “program.” The search strategies were as follows: “older adults, sleep, therapy,” “older adults, sleep, intervention,” “older adults, sleep, program,” “older adults, insomnia, therapy,” “older adults, insomnia, intervention,” and “older adults, insomnia, program.”

### 2.3. Study Selection

Studies were selected/excluded as per the PRISMA 2020 protocol [31]. Studies were classified using RefWorks (ProQuest, Ann Arbor, MI, USA). Duplicate search results were removed automatically. Titles and abstracts were reviewed to determine the eligibility per our inclusion/exclusion criteria. The full texts of the articles were then reviewed. Two researchers independently performed the literature search and resolved disagreements arising during the review process by consensus.

### 2.4. Data Extraction

An analytical framework for reviewing the included studies was developed based on the PICO-SD strategy. Two researchers independently extracted the study contents and compared the results. The analytical framework included the first author (year), country, participants (number, sex, mean age/age range, and participant characteristics), interventions (type, contents, number/timing/duration of sessions, setting, and controls), and outcomes (sleep measures, sleep outcomes, and other measures).

### 2.5. Quality Assessment

The risk of bias (RoB) of the included RCTs was evaluated using the revised Cochrane RoB tool for randomized trials (RoB2, short version) [33]. The RoB2 evaluates RoB arising from randomization, deviations from the intended interventions, effect of assignment to interventions, missing outcome data, outcome measurement, and selection of reported results, as well as overall RoB [33]. The signaling questions for each domain were evaluated based on response options. RoB judgments (low risk, some concerns, and high risk) were created for each domain as per the algorithm for the suggested judgment of RoB arising from the randomization process [33].

Each researcher performed quality appraisals while collating studies using the RoB2 template. Disagreements regarding quality appraisal were resolved by discussion. Quality ratings were entered into the RevMan 5.4 program (Cochrane Training, London, UK).

RCTs were evaluated using the Physiotherapy Evidence Database (PEDro) scale [34]. The PEDro scale is an 11-item scale designed for rating the methodological quality of RCTs. The PEDro criteria are: specification of eligibility criteria, random allocation, concealed allocation, prognostic similarity at baseline, subject blinding, therapist blinding, assessor blinding, >85% follow-up for at least one key outcome, intention-to-treat analysis, between group statistical comparison for at least one key outcome, and point estimates and measures of variability provided for at least one key outcome [34]. Each item satisfied (except for item 1) contributes one point to the total PEDro score (range = 0–10 points) [34]. A trial was considered to be of high quality if it received a score of at least 6/11 on the PEDro quality scale [35].

### 2.6. Meta-Analysis

We performed a meta-analysis evaluating the effects of non-pharmacological sleep interventions in older adults using R statistical software (v.4.1.3, The R Project for Statistical Computing, Vienna, Austria). An analytical framework was developed, and the mean/standard deviation (SD) and number of participants at baseline and post-intervention tests were coded using Excel (Microsoft, Seattle, WA, USA). We excluded studies that did not clearly present results as baseline and post-intervention values. For the Pittsburgh Sleep Quality Index (PSQI) and Insomnia Severity Index (ISI), higher scores indicate worse sleep; hence, the results were reverse-coded for analysis [32]. Only the CBT for insomnia (CBT-I) standard group score (not the CBT-I+ advanced group score) was included.

A random effects model was applied under the assumption of heterogeneity of study populations. Effect sizes were examined visually using a forest plot. Due to the small number of studies available for each intervention, only the overall effectiveness of non-pharmacological sleep interventions was evaluated. Homogeneity was tested using I^2^ and Q values, where I^2^ ≤ 25%, 50%, and ≥75% indicated low, medium, and high heterogeneity, respectively [36]. To examine the heterogeneity of non-pharmacological interventions among the studies, a meta-ANOVA was conducted. Publication bias was assessed by checking for left-right symmetry of the funnel plot.

### 2.7. Ethical Considerations

This study was exempted from review by the Institutional Review Board at the authors’ affiliated institution (KSU-22-04-001). As this study did not collect or record personally identifiable information of study subjects and used only previously generated data or documents, exemption from review was obtained.

## 3. Results

### 3.1. Literature Selection

The selection process is shown in Figure 1. A total of 2881 search results were generated; 2361 studies were non-Korean (PubMed, n = 949; MEDLINE, n = 875; Cochrane CENTRAL, n = 331; and CINAHL, n = 206) and 520 were Korean (NDSL, n = 73; KISS, n = 56; RISS, n = 254; and DBpia, n = 137). RefWorks automatically removed 592 duplicates; 2068 studies were removed after reviewing titles and abstracts.

The full text was reviewed for the 221 remaining studies; 206 articles not meeting the inclusion criteria were removed (age criteria, n = 118; non-pharmacological interventions, n = 2; no control group, n = 13; not presenting sleep-related outcomes, n = 6; incorrect study type, n = 59; no full text, n = 7; and non-English/non-Korean publications, n = 1). As a result, 15 studies were systematically reviewed (Table 1). Five studies not providing post-intervention measurements or not clearly presenting means/SDs were removed, resulting in a total of 10 studies included in the meta-analysis.

### 3.2. Studies and Interventions

As mentioned above, 15 studies (n = 913 participants) were systematically reviewed (Table 1) (Republic of Korea, n = 4; Turkey, n = 3; Japan, n = 2; Taiwan, n = 2; US, Spain, China, and Australia, n = 1 each). Both sexes were included in 14 studies, while one study only included women as it was conducted among breast cancer patients. The mean age was ≥75 years in nine studies and <75 years in six studies.

The most common non-pharmacological sleep intervention evaluated in older adults was exercise (n = 7), followed by aromatherapy (n = 3), auricular acupressure therapy (n = 2), CBT (n = 2), and meditation (n = 1). The detailed sleep intervention characteristics are described in Table 1.

Two studies employed a 4-week exercise program, while other exercise studies evaluated 8-week, 10-week, 20-week, 6-month, and 12-month durations (n = 1 each)**.** Two studies evaluated a 4-week aromatherapy program and one study evaluated a 24-week program. Auricular acupressure therapy was performed for 6 or 8 weeks (n = 1 each). CBT was performed for 8 weeks in one study; the duration was not specified in another study. Meditation was performed for 8 weeks in one study.

The most common frequency of exercise therapy sessions was three times/week (n = 4 studies). One study employed a progressively increasing number of sessions (three, four, and seven days/week). In another study, an exercise program was applied once a week, while the evaluated program was applied three times/day in one study. Aromatherapy was performed every day in two studies and three times/week in one study. In two studies evaluating auricular acupressure, intervention was performed once a week; one session consisted of five consecutive intervention days followed by two days of rest. CBT was applied six times in one study and eight times in another study. Meditation was applied every day in one study.

Two studies each evaluated 40 min and 60 min exercise therapy sessions (other exercise therapy durations included 3 min, 30 min, and 80 min sessions, n = 1 each). Five minute and ≥10 min sessions were evaluated for aromatherapy interventions (n = 1 study each); one aromatherapy study did not provide information on session duration. Session duration was not specified for auricular acupressure. CBT was performed for ≥60 min and meditation for 45 min.

We noted that nine studies did not provide the experimental intervention to the control group, but six studies provided a comparative intervention.

In regard to sleep quality, non-pharmacological sleep interventions were most commonly evaluated using the PSQI for sleep quality (n = 8); one study used the visual analog scale (VAS) [48]. Sleep satisfaction was assessed using the Verran and Snyder–Halpern (VSH) Sleep Scale [44,45] (n = 2), and for insomnia, the ISI was used [38,46] (n = 2).

Objective sleep features were assessed using a bioactive device (n = 4) or serum melatonin levels (n = 1). Degree of sleep was assessed using the Oviedo Sleep Questionnaire [43] (n = 1) and sleep journals (n = 1). A total of 14 studies reported that non-pharmacological sleep interventions were effective in older adults [14,15,37,38,39,40,41,43,44,45,46,47,48,49]. One study reported that the effectiveness of the intervention was not confirmed [42].

### 3.3. Literature Quality

Of the 15 studies included in the systematic review, five were excluded due to not clearly presenting post-intervention measurements [38,41,42,47,48]. Ten studies were evaluated for quality. Figure 2 shows the results of the RoB evaluations for these studies.

In regard to the randomization process, seven studies were given a “low risk” rating. Three studies were given a “some concerns” rating. In regard to deviation from the intended intervention, eight studies were assessed as “low risk”; two studies were assessed as presenting “some concerns.” In regard to missing outcome data, all studies were assessed as “low risk.” In regard to outcome measurement, two studies were assessed as “low risk” and eight studies were assessed as presenting “some concerns.” In regard to selection of the reported results, all 10 studies were assessed as “low risk.”

In some studies, rated as having “some concerns,” it was unclear whether the researchers had knowledge about the intervention. Since no studies were assessed as “high risk,” all 10 studies were included in the meta-analysis.

Table 2 shows the results of the PEDro scale evaluations for ten studies. All studies had a quality level of 6 points or higher, and 2 studies showed a very high quality level of 9 points.

### 3.4. Effectiveness of Sleep Interventions

Figure 3 shows the results of the meta-analysis evaluating non-pharmacological sleep interventions in older adults. We evaluated exercise therapy (n = 4), aromatherapy (n = 2), auricular acupressure (n = 2), CBT (n = 1), and meditation interventions (n = 1).

Two studies that applied auricular acupressure presented the mean/SD for deep sleep duration [14,15] and melatonin concentrations [15] in addition to sleep quality; these measurements were also included in the meta-analysis. We analyzed 13 outcome measures.

The effects of non-pharmacological sleep interventions in the studies included in the meta-analysis (n = 712 participants) were as follows (Figure 3). The estimated average effect size (Hedge’s g) was 1.00 (*p* = 0.020) in the random effects model, showing that older adults’ sleep increased by an SD of 1.00 following non-pharmacological intervention. The confidence interval (CI) did not include 0 (95% CI 0.16, 1.85), and the null hypothesis of no effect was rejected (Z = 2.33, *p* = 0.020). Thus, the evaluated non-pharmacological sleep interventions were deemed effective in older adults. There was a high degree of heterogeneity in the included studies (I^2^ = 92%, Q = 155.4, df = 12, *p* < 0.001).

### 3.5. Analysis of the Moderating Effect

In this study, the overall heterogeneity of non-pharmacological interventions was found to be I^2^ = 92% (Q = 155.4, *p* < 0.001), indicating that a large proportion of the effect size variation among studies was associated with sex, average age, and intervention method. A meta-ANOVA was conducted, with measurement tools and intervention period as moderating variables. As a result of analyzing sex, age, intervention method, measurement tool, and intervention period as moderating variables, the difference in effect size among groups was not statistically significant. Details are described in Table 3.

### 3.6. Publication Bias

A funnel plot was generated to examine publication bias; asymmetry was observed (Figure 4).

After removing five outliers, I^2^ dropped from 92% (Q = 155.4, df = 12, *p* < 0.001) to I^2^ = 17% (Q = 8.40, df = 7, *p* = 0.298). The average effect size (Hedge’s g) in the random effects model decreased from 1.00 (95% CI 0.16, 1.85) to 0.70 (95% CI 0.47, 0.93), but statistical significance was maintained (Figure 5).

## 4. Discussions

We comprehensively analyzed the characteristics of non-pharmacological sleep interventions in older adults and evaluated their effectiveness through a systematic review and meta-analysis.

The mean age of the participants included in the systematic review was ≥75 years in nine studies and <75 years in six studies. Fourteen studies (93%) were conducted in both men and women. Most studies had a larger female study population. A Statistics Korea report [50] showed the following distribution for the older adult population in the Republic of Korea: ages 65–69 years (34%), 70–74 years (25%), and ≥75 years (42%), with more females (57%) than males (44%). Overall, this suggests that older adults aged ≥ 75 years and older women require more attention in future research endeavors.

The older adults included in this analysis had cognitive impairment, depression, insomnia, sleep disorders, and/or other physical problems. The average proportion of the population that is classified as older adults in OECD countries is 17.64% (2021); countries with higher proportions than the OECD average are Japan (28.86%), Italy (23.67%), Greece (22.69%), Portugal (22.55%), Germany (22.06%), France (20.85%), the United Kingdom (18.83%), and Canada (18.52%) [51]. According to the 2021 Elderly Statistics report by Statistics Korea [50] the leading cause of death per 100,000 population in older adults aged ≥65 years is cancer, followed by heart disease, pneumonia, cerebrovascular disease, and Alzheimer’s disease. This indicates that more attention should be paid to sleep in older adults suffering from diseases with high mortality rates.

As shown here, given the clear sex ratio difference and varying health statuses in the older population, we anticipated that sleep outcomes would differ according to these factors. The evaluated studies did not consistently present such secondary analyses, which hindered subgroup analyses. Future sleep intervention studies in older adults should concentrate on the health of older women and should examine outcomes of sleep interventions by sex and health status.

The most common non-pharmacological sleep intervention evaluated in our review was exercise, followed by aromatherapy, auricular acupressure, CBT, and meditation. Exercise regimens varied widely, including Healthy Beat Acupunch exercises, barefoot walking, lip closure training, and yoga. Other previous studies evaluated Pilates [12,52], forest walking programs [53], and Tai Chi [54]. Among them, a recent study indicated that Tai Chi was effective in improving older adults’ sleep quality [54], but the study was not included in the current analysis because it did not meet the selection criteria.

Aromatherapies included inhalation therapy, bathing therapy, and hand massage. In two studies, lavender oil was used, while aromatherapy oil was used in another study. In addition to the method of using the aroma oil included in this study, in which 10% damask rose was placed inside a pillow before going to bed and the scent was inhaled [13], participants in another study were asked to continuously wear a pendant containing marjoram and orange oil [55].

In both studies that evaluated auricular acupressure, a tape containing *Melandryum firmum* seeds was applied to each participant’s ears. Chen et al. [56] demonstrated that acupressure and valerian essential oil applied to the inner wrists and feet were effective in improving sleep in intensive care unit patients. Studies evaluating CBT were conducted in patients with cognitive impairment, depression, or insomnia; the methods were similar to those of previous investigations [16,17].

Meditation was performed at home using audio-recorded meditation instructions. However, Huberty et al. [57] attempted to provide easier access to meditation by using the “Calm” smartphone app.

Other studies evaluated exergaming [58], music therapy [59], and animal-assisted therapy [60], but these studies did not meet the inclusion criteria for this review. More experimental studies are needed to develop and evaluate cost-effective and easy-to-administer non-pharmacological sleep interventions in older adults that are appropriate for the evolving digital social environment and tailored according to individual health status.

In this review, the most common duration of non-pharmacological sleep interventions was ≥10 weeks (n = 5 studies; exercise therapy and aromatherapy). One study of the administration of an auricular acupressure intervention used a 6-week program, and one study each of exercise therapy, auricular acupressure, CBT, and meditation used 8-week programs. The shortest duration intervention (4 weeks) was employed for exercise therapy and aromatherapy. The duration of the intervention varied widely for non-pharmacological sleep interventions and differed substantially even for the same type of intervention.

We note that some studies did not assess/present post-test measurements. In addition, some studies conducted a 6-month or 12-month follow-up, although the follow-up periods were shorter in Korean studies. In future, experimental studies should adequately present post-test results, and should evaluate the long-term effects of non-pharmacological sleep interventions in consideration of the unique physical, mental, and social characteristics of the older population.

Most studies used the PSQI to assess sleep quality; the VAS was also used. Other evaluated sleep outcomes included sleep satisfaction, insomnia, and degree of sleep. Although some studies measured deep sleep duration using a bioactive device or measured blood melatonin levels, most studies assessed subjective outcomes using questionnaires. In recent years, technological advances have enabled the use of noninvasive and efficient means, including the Dayzz app [61], the Personal Activity Intelligence e-Health program [62], the Calm mindfulness meditation app [18], and wearable devices [20]. Future experimental studies should employ objective sleep outcomes in addition to subjective assessments tailored to the characteristics of older adults.

Our meta-analysis demonstrated that non-pharmacological sleep interventions are highly effective in older adults. However, the included studies showed substantial heterogeneity. Therefore, we checked for publication bias. Homogeneity was achieved after removing five outlier studies. Although the effect size was reduced to medium, the associated CI remained statistically significant. Previous meta-analyses of studies evaluating various non-pharmacological sleep interventions in older adults, including exercise [27,28], aromatherapy [25,26], meditation [29], and CBT [9] also reported the efficacy of non-pharmacological sleep interventions. However, many older adults still suffer from sleep problems. Non-pharmacological sleep interventions should be more actively utilized, not only in clinical practice, but also in communities as a means to address sleep problems in this demographic.

Our study confirmed the efficacy of non-pharmacological sleep interventions in older adults via a systematic review and meta-analysis. Our study had the following limitations. First, the older adult population features clear and distinct differences in sex ratio and health status compared to younger adults, which may impact outcome measurements. The included studies did not uniformly present these characteristics and we could not perform subgroup analyses. Second, five studies included in the systematic review were excluded from the meta-analysis because they did not measure or incompletely reported the post-test measurements. Third, we could not analyze the effectiveness of each type of intervention due to the small number of relevant studies. Fourth, the keyword, exercise therapy, may not include all types of exercises such as Tai Chi, Qigong, and dance. Fifth, this review was not registered and a protocol was not prepared. However, our study contributes to providing a comprehensive understanding of non-pharmacological sleep interventions in older adults and confirmed that non-pharmacological interventions are effective in improving sleep in this population. We presented scientific evidence for developing sleep intervention guidelines in older adults.

## 5. Conclusions

Our systematic review and meta-analysis demonstrated the efficacy of non-pharmacological sleep interventions in older adults. Non-pharmacological sleep interventions evaluated included exercise, aromatherapy, auricular acupressure, CBT, and meditation. The main sleep outcome was self-reported sleep quality.

Non-pharmacological interventions should be actively utilized in clinical practice and community settings to improve sleep problems in this demographic. Future studies should investigate the health of older women and the long-term effects of non-pharmacological sleep interventions. Objective measures should be used to assess the effectiveness of these interventions.

## Figures and Tables

**Figure 1 ijerph-20-03101-f001:**
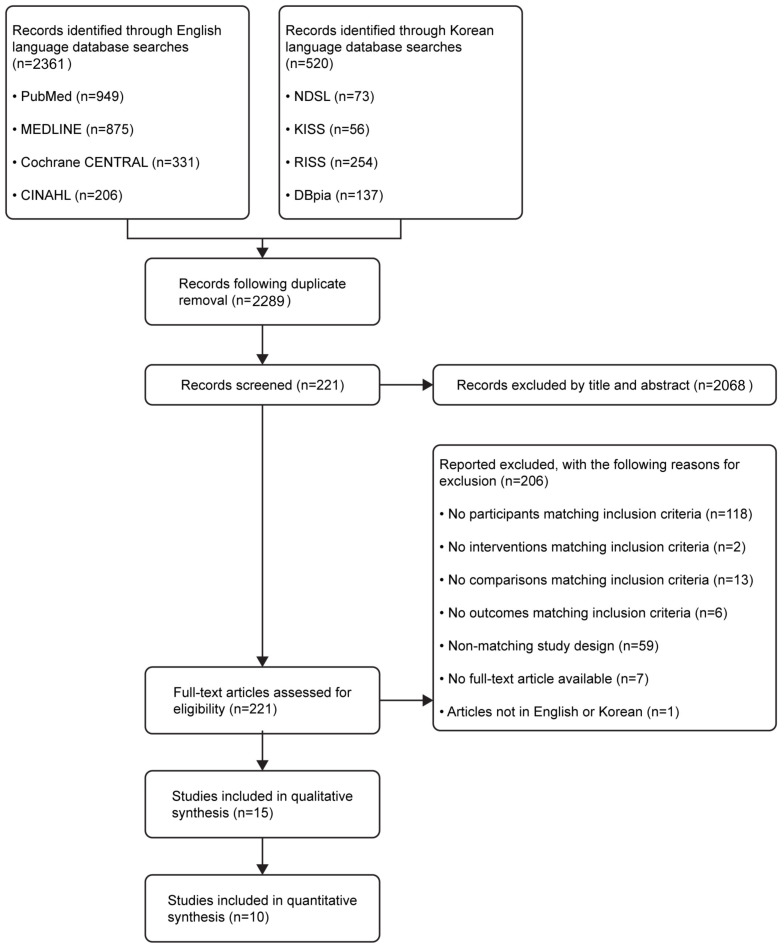
Flow diagram illustrating the study selection process.

**Figure 2 ijerph-20-03101-f002:**
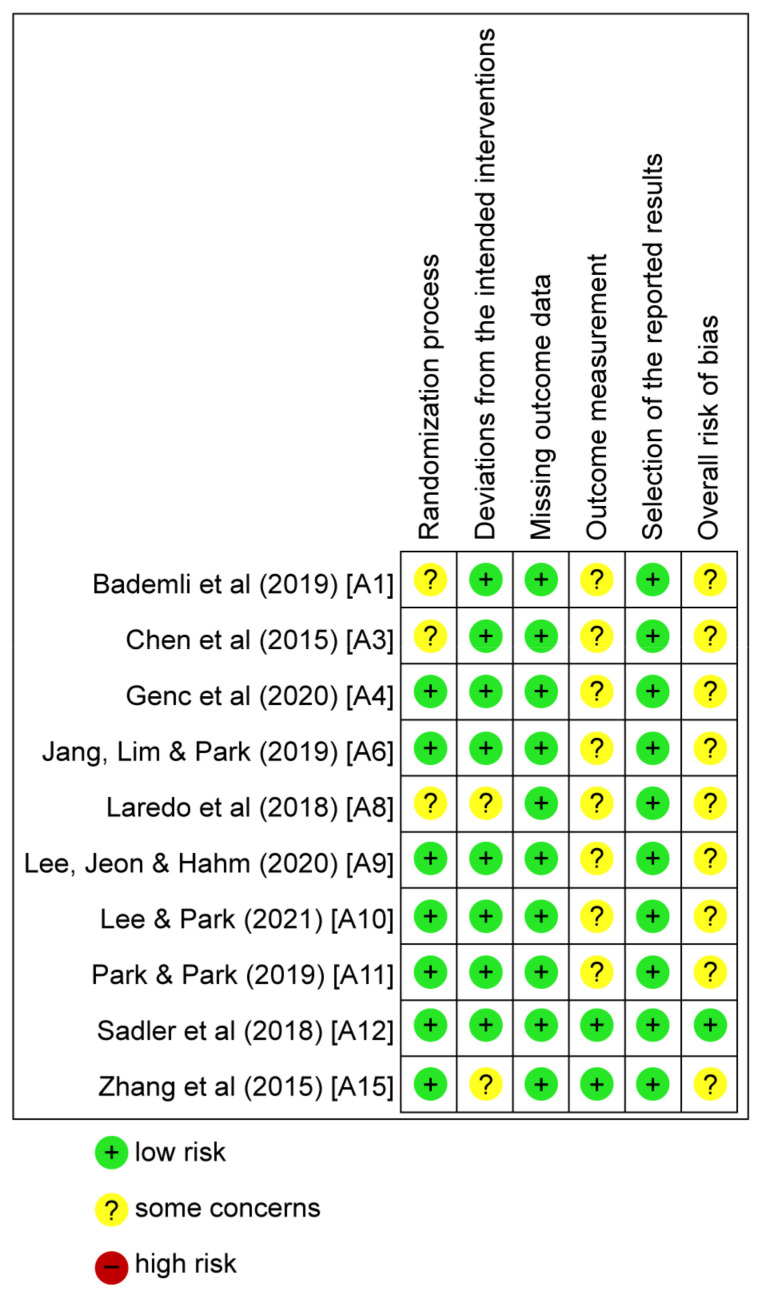
Risk of bias graph. Green represents low risk, yellow represents some concerns, and red represents high risk [14,15,37,39,40,43,44,45,46,49].

**Figure 3 ijerph-20-03101-f003:**
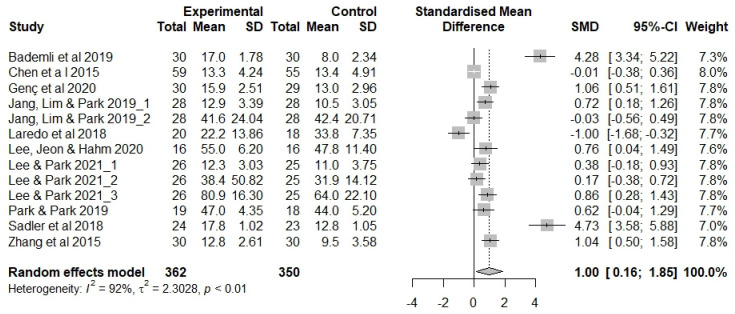
Forest plot showing the effects of non-pharmacological interventions. SD, standard deviation; SMD, standard mean difference [14,15,37,39,40,43,44,45,46,49].

**Figure 4 ijerph-20-03101-f004:**
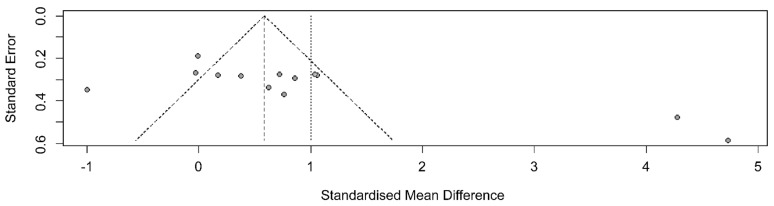
Funnel plot showing the effects of non-pharmacological interventions.

**Figure 5 ijerph-20-03101-f005:**
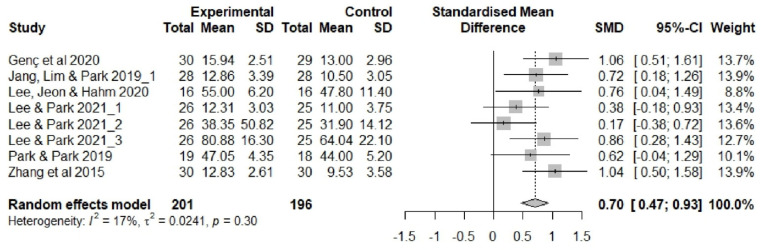
Forest plot showing the effects of non-pharmacological interventions (modified). SD, standard deviation; SMD, standard mean difference [14,15,40,44,45,49].

**Table 1 ijerph-20-03101-t001:** Systematic review evaluating the effects of non-pharmacological interventions on sleep in older adults (n = 15).

Author(s) (Year) (No.)	Country	Participants			Intervention					Outcomes		
		Number (n or % by Sex)	Mean Age or Range	Characteristics	Type	Contents	No. of Sessions/ Time of Session/Duration	Setting	Control Group	Sleep Measure	Sleep Outcome	Other Measures
Bademli et al. (2019) [37] (A1)	Turkey	Exp.: 30 (M: 12/F: 18) Con.: 30 (M: 13/F: 17)	Exp.: 72.2 Con.: 70.7	Older adults with mild cognitive impairment	Exercise	Physical activity program	Three, four, or seven days per week/80 min/20 weeks	Nursing home	N/A	PSQI	Effective	SMMSE
Cassidy-Eagle et al. (2018) [38] (A2)	USA	Exp.: 14 Con.: 13 (M < F)	Exp.: 89.4 Con.: 88.7	Older adults with mild cognitive impairment and insomnia	Cognitive behavior therapy	Sleep hygiene, relaxation, sleep scheduling, and cognitive therapy	Six sessions/60 min/-	Two local independent and assisted living facilities	N/A	ISI sleep measure (ActiGraph, wGT3x)	Effective	HVLT-R, D-KEFS, ADL, IADL, MoCA
Chen et al. (2015) [39] (A3)	Taiwan	Exp.: 59 Con.: 55 (M: 50.9%/F: 49.1%)	Mean: 79.2	Older adults in wheelchairs	Exercise	Wheelchair-bound senior elastic band exercise program	Three times per week/40 min/six months	Nursing home	N/A	Chinese PSQI	Effective	TDQ
Genç et al. (2020) [40] (A4)	Turkey	Exp.: 30 (M: 22/F: 8) Con.: 29 (M: 25/F: 4)	Exp.: 74.5 Con.: 72.0	Older adults	Aromatherapy	Aroma inhalation (lavender oil)	Every day/-/one month	Nursing home	N/A	PSQI	Effective	FSS
Hsiao et al. (2018) [41] (A5)	Taiwan	Exp.: 113 (M: 20/F: 93) Con.: 119 (M: 30/F: 89)	Exp.: 74.7 Con.: 73.9	Older adults	Exercise	Healthy Beat Acupunch exercises	Three times per week/40 min/12 months	Community care centers	N/A	PSQI	Effective	SF-12
Jang, Lim, and Park (2019) [14] (A6)	Republic of Korea	Exp.: 28 (M: 0/F: 28) Con.: 28 (M: 2/F: 26)	Exp.: 79.2 Con.: 78.4	Older adults with knee osteoarthritis	Acupressure	Auricular acupressure (Shen Men, knee liver, heart occiput)	Eight sessions/-/eight weeks	Elderly welfare facility	Auricular acupressure (Helix five-point)	Korean PSQI, wrist sleep device (Fitbit alta HR)	Effective	VAS, PPTs Korean WOMAC, knee flexion, knee extension
Kouzuki et al. (2020) [42] (A7)	Japan	0.1%: 13 (M: 9/F: 4) 0.5%: 11 (M: 4/F: 7) 1.0%: 11 (M: 4/F: 7)	0.1%: 76.0 0.5%: 78.0 1.0%: 82.0	Older adults with Alzheimer’s disease, mild cognitive impairment	Aromatherapy	Bath salt containing an aroma component (0.1%)	Every day/≥ 10 min/24 weeks	Hospital outpatient clinic	Bath salt containing an aroma component (0.5%, 1.0%)	Japanese PSQI	Not effective	TDAS, OSIT-J
Laredo et al. (2018) [43] (A8)	Spain	Exp.: 20 (M: 22.2%/F: 77.8%) Con.: 18 (M: 28.5%/F: 71.5%)	Exp.: 75.4 Con.: 76.4	Healthy older adults	Exercise	10-week functional training program	Three times per week/60 min/10 weeks	Community in Loja, Granada	N/A	OSQ	Effective	VAS, PMS, GDS
Lee, Jeon, and Hahm (2020) [44] (A9)	Republic of Korea	Exp.: 16 (M: 1/F: 15) Con.: 16 (M: 1/F: 15)	Exp.: 76.6 Con.: 76.1	Older adults with low back pain	Exercise	Barefoot walking	Three times per week/30 min/four weeks	Sandy beach in community	Sneaker walking	VSH	Effective	VAS, ODI, Berg balance scale, timed UP and Go test, Korean QoL
Lee and Park (2021) [15] (A10)	Republic of Korea	Exp.: 26 (M: 13/F: 13) Con.: 25 (M: 12/F: 13)	Exp.: 74.8 Con.: 72.6	Older adults with sleep disorders	Acupressure	Auricular acupressure (Shen Men, heart, anterior occipital lobe)	Six sessions/-/six weeks	Elderly welfare facility	Auricular acupressure (helix four-point)	Korean PSQI, wrist sleep device (Fitbit Charge HR) blood melatonin levels	Effective	N/A
Park and Park (2019) [45] (A11)	Republic of Korea	Exp.: 19 (M: 4/F: 15) Con.: 18 (M: 4/F: 14)	Exp.: <75, n = 9; ≥75, n = 10 Con.: <75, n = 5; ≥75 n = 13	Older adults long-term care facility residents	Aromatherapy	Hand massage with preferred aroma oil	Three times per week/5 min/four weeks	Long-term care facility	Hand massage with lavender oil	Korean VSH	Effective	PSQ
Sadler et al. (2018) [46] (A12)	Australia	CBT-I: 24 (M: 9/F: 15) CBT-I+: 25 (M: 12/F: 13) Con.: 23 (M: 10/F: 12/transgender: 1)	Exp.: 74.7 Con.: 72.3	Older adults with insomnia and depression	Cognitive behavior therapy	Combination of educational, cognitive, and behavioral interventions	CBT-I (standard): eight sessions/60~90 min/eight weeks CBT-I + (advanced): eight sessions/75~90 min/eight weeks	Case managed by an aged persons’ community mental health service	N/A	ISI, CDS, DBAS-10, and SLEEP-50 scale	Effective	GDS, GAI-SF, BHS, EQ-5D-3L
Takamoto et al. (2018) [47] (A13)	Japan	Exp.: 10 (M: 2/F: 8) Con.: 10 (M: 0/F: 10)	Exp.: 87.3 Con: 85.3	Older adults	Exercise	Lip closure training	Three times per day/3 min/four weeks	Elder care facility	N/A	Upper arm sleep device (ActiSleep Monitor)	Effective	LIP DE CUM, Everio GZ-MG275, NIRS
Yagli and Ulger (2015) [48] (A14)	Turkey	Exp.: 10 (F: 10) Con.: 10 (F: 10)	Exp.: 68.6 Con.: 68.9	Older adults with breast cancer patients	Exercise	Yoga	Eight sessions/60 min/eight weeks	Department of Physiotherapy and Rehabilitation	Exercise program	VAS	Effective	Turkish-NHP, Turkish-BDI, VAS
Zhang et al. (2015) [49] (A15)	China	Exp.: 30 (M: 16/F: 14) Con.: 30 (M: 19/F: 11)	Exp.: 78.6 Con.: 77.6	Older adults with chronic insomnia	Meditation	Mindfulness- based stress reduction	Every day/45 min/eight weeks	Medical Psychology Division of the general hospital	N/A	PSQI	Effective	SAS, GDS

ActiGraph wGT3x (ActiGraph Corporation, Pensacola, FL, USA): to monitor subjective and objective sleep; ActiSleep Monitor (Actigraph, Pensacola, USA): to measure circadian rest-activity rhythm; ADL: activities of daily living; BDI: Beck Depression Inventory; BHS: Beck Hopelessness Scale; CBT-I: cognitive behavior therapy for insomnia; CDS: Consensus Sleep Diary; Con.: control group; DBAS-10: Dysfunctional Beliefs and Attitudes About Sleep 10-Item Scale; D-KEFS: Delis-Kaplan Executive Function System; EQ-5D-3L: EuroQol 5-D 3-L Scale; Everio GZ-MG275 (Victor, Kanagawa, Japan): to record eating behavior; Exp.: experimental group; F: female; Fitbit Alta HR (Fitbit, China): to monitor sleep phase; Fitbit Charge HR (FitBit Inc, San Francisco, CA, USA): to monitor objective quality and quantity of sleep; FSS: Fatigue Severity Scale; GAI-SF: Geriatric Anxiety Inventory-Short Form; GDS: Geriatric Depression Scale; HVLT-R: Hopkins Verbal Learning Test-Revised; IADL: instrumental activities of daily living; ISI: Insomnia Severity Index; Korean QoL: Korean version of WHO Quality of Life Scale; LIP DE CUM (Cosmos Instruments Co. Ltd., Tokyo, Japan): to measure maximal lip closure force; M: male; MoCA: Montreal Cognitive Assessment; N/A: not available; NHP: Nottingham Health Profile; NIRS: near-infrared spectroscopy; No.: number; ODI: Oswestry Disability Index; OSIT-J: Odor Stick Identification Test for Japanese; OSQ: Oviedo Sleep Questionnaire; POMS: Profile of Mood States; PPTs: pressure pain thresholds; PSQ: Perceived Stress Questionnaire; PSQI: Pittsburgh Sleep Quality Index; SAS: Zung Self-Rating Anxiety Scale; SF-12: 12-Item Short Form Health Survey; SMMSE: Standardized Mini-Mental State Examination; TDAS: Touch Panel-type Dementia Assessment Scale; TDQ: Taiwanese Depression Questionnaire; VAS: visual analog scale; VSH: Verran and Snyder-Halpern Sleep Scale; WOMAC: Western Ontario and McMaster Universities Osteoarthritis Index.3.3. Literature Quality.

**Table 2 ijerph-20-03101-t002:** Methodological scores assigned to the RCTs.

	PEDro Criterion												
Author(s) (Year)	(1)	(2)	(3)	(4)	(5)	(6)	(7)	(8)	(9)	(10)	(11)	Total	Quality
Bademli et al. (2019) [37]	1	1	1	1	0	0	0	1	1	1	1	7	Good
Chen et al. (2015) [39]	1	1	1	1	0	0	0	1	1	1	1	7	Good
Genç et al. (2020) [40]	1	1	1	1	0	0	0	1	1	1	1	7	Good
Jang, Lim, and Park (2019) [14]	1	1	1	1	0	0	0	1	1	1	1	7	Good
Laredo et al. (2018) [43]	1	1	0	1	0	0	0	1	1	1	1	6	Good
Lee, Jeon, and Hahm (2020) [44]	1	1	1	1	0	0	0	1	1	1	1	7	Good
Lee and Park (2021) [15]	1	1	1	1	0	0	0	0	1	1	1	6	Good
Park and Park (2019) [45]	1	1	1	1	0	0	0	1	1	1	1	7	Good
Sadler et al. (2018) [46]	1	1	1	1	1	1	0	1	1	1	1	9	Excellent
Zhang et al. (2015) [49]	1	1	1	1	1	1	0	1	1	1	1	9	Excellent

Item score 0 = absent, 1 = present. The PEDro criteria are: (1) specification of eligibility criteria; (2) random allocation; (3) concealed allocation; (4) prognostic similarity at baseline; (5) subject blinding; (6) therapist blinding; (7) assessor blinding; (8) > 85% follow-up for at least one key outcome; (9) intention-to-treat analysis; (10) between group statistical comparison for at least one key outcome; (11) point estimates and measures of variability provided for at least one key outcome.

**Table 3 ijerph-20-03101-t003:** Effects of moderator variables.

Category	Subgroup	k	Hedge’s g	95% CI		Z (*p*)	I^2^	Q (*p*)
				Lower Limit	Upper Limit			
Sex	Male	3	0.03	−0.06	0.12	0.63 (0.530)	0%	0.60 (0.740)
	Female	10	0.09	−0.29	0.47	0.46 (0.648)		
Age	<75	6	0.09	−0.05	0.21	1.17 (0.243)	0%	1.37 (0.505)
	≥75	7	−0.00	−0.12	0.11	−0.06 (0.955)		
Intervention type	Acupressure therapy	5	0.04	−0.08	0.17	0.68 (0.498)	0%	1.27 (0.945)
	Aromatherapy	2	0.02	−0.19	0.22	0.16 (0.873)		
	Cognitive behavioral therapy	1	0.17	−0.39	0.74	0.61 (0.545)		
	Exercise	4	0.00	−0.16	0.16	0.02 (0.982)		
	Meditation	1	0.22	−0.506	0.94	0.59 (0.553)		
Intervention period	4 wks	3	0.03	−0.11	0.17	0.36 (0.717)	0%	1.14 (0.888)
	6 wks	3	0.05	−0.09	0.20	0.72 (0.470)		
	8 wks	4	0.06	0.16	0.28	0.51 (0.612)		
	≥10wks	3	−0.07	−0.36	0.22	−0.48 (0.631)		
Sleep measurement	Blood sample	1	0.06	−0.24	0.37	0.39 (0.695)	0%	3.53 (0.741)
	Fitbit	2	0.04	−0.11	0.18	0.48 (0.634)		
	ISI	1	0.17	−0.39	0.74	0.61 (0.545)		
	Oviedo.S.Q	1	−0.21	−0.56	0.13	−1.21 (0.228)		
	PSQI	6	0.16	−0.12	0.43	1.12 (0.264)		
	VSH	2	0.02	−0.12	0.17	0.28 (0.777)		

## Data Availability

Not applicable.

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
