# Peer review of "Effects of Non-Pharmacological Sleep Interventions in Older Adults: A Systematic Review and Meta-Analysis"

_ijerph, 2023, doi:10.3390/ijerph20043101_

Round 1
Reviewer 1 Report
1. The authors have to describe the differences among this research and reference 24, reference 32. (lines 54-56)
2. The inclusion criteria for the participants need to be clarified. (lines 78-79)
3. Please clarify what is the qualitative synthesis in Figure 1.
Reviewer 2 Report
The authors evaluated the effects of non-pharmacological sleep interventions in older adults. They identified the need to conduct this study. However, I have some suggestions to improve the quality of this review.
1. The search might not be comprehensive enough. As indicated by Leung et al.(2022), Tai Chi can improve the sleep quality among older adults. But Tai Chi was not identified in the included study. The authors might need to refine their search.
2. Section 3.2, regarding the characteristics of studies and interventions (line 205-225) can be summarized into shorter paragraphs, and the details can be referred to Table 1.
3. Section 3.4, it was unclear why there were 13 outcomes in 10 studies.
4. Section 3.5, please describe the identification of outliners. Also, the outliners should be cited in the text for reader to identify easily.
5. Please show the name of authors and year of the studies in Figures 3&4. The readers cannot identify the values extracted from which studies by numbering.
6. Section 4, discussion, line 49-61 needed to be revised. As this is an international journal, the authors should not focus data from Korea only. Alternatively, the global trend should be taken consideration.
7. Section 4, discussion, line 62 indicated sex might be an influencing factor on sleep quality. How about the influence of age? The authors indicated there were nine studies with participants aged over 75 years and six studies with participants aged < 75 years. The authors might be able to conduct subgroup analysis regarding different age groups.
8. The authors identified seven studies used exercise as non-pharmacological intervention to improve sleep quality. Please conduct a meta-analysis to examine the effect of exercise on sleep quality and compare to the effect of Tai Chi (Leung et al., 2022).
9. "Aromatherapies included inhalation therapy....marjoram and orange oil [51]." What is the purpose of this paragraph? The paragraph had no connection with previous paragraph.
10. "Meditation was performed at home using .... according to individual health status." What is purpose of this paragraph? What are the findings of these studies [53-56]?
11. It is unclear the measures of blood pressure, body mass index, hemoglobin, and serum creatinine can reflect the sleep quality.
Reference:
Leung LYL, Tam HL, Ho JKM. Effectiveness of Tai Chi on older adults: A systematic review of systematic reviews with re-meta-analysis. Arch Gerontol Geriatr. 2022;103:104796. doi:10.1016/j.archger.2022.104796
Reviewer 3 Report
First of all, congratulations for the research work done, then I will mention only two recommendations in order to get a clearer and more precise information of your results. As far as methodology is concerned, I have to congratulate you on your work.
The first one is to apply the pedro scale methodological quality analysis. Together with the cochrane analysis already applied, it would add methodological quality.
The second is to inform if the search protocol was registered in any platform such as PROSPERO and if not, to mention it as a limitation of the study. This is essential to avoid bias during the study.
Congratulations for your work
Round 2
Reviewer 2 Report
The revised manuscript is good as the authors addressed most of my concern appropriately. However, I still have a concern regarding the keywords used in the search.
Since this review focused on non-pharmacological intervention, I am not sure the keyword (exercise therapy) can cover all types of exercise such as Tai Chi, Qigong, dancing, etc.
